# Clinical Routine *TERT* Promoter Mutational Screening of Follicular Thyroid Tumors of Uncertain Malignant Potential (FT-UMPs): A Useful Predictor of Metastatic Disease

**DOI:** 10.3390/cancers11101443

**Published:** 2019-09-26

**Authors:** Martin Hysek, Johan O. Paulsson, Kenbugul Jatta, Ivan Shabo, Adam Stenman, Anders Höög, Catharina Larsson, Jan Zedenius, Carl Christofer Juhlin

**Affiliations:** 1Department of Oncology-Pathology, Karolinska Institutet, 17176 Stockholm, Sweden; martin.hysek@ki.se (M.H.); johan.paulsson@ki.se (J.O.P.); adam.stenman@ki.se (A.S.); anders.hoog@ki.se (A.H.); catharina.larsson@ki.se (C.L.); 2Department of Pathology and Cytology, Karolinska University Hospital, 17176 Stockholm, Sweden; kenbugul.jatta@sll.se; 3Department of Breast, Endocrine Tumors and Sarcoma, Karolinska University Hospital, 17176 Stockholm, Sweden; ivan.shabo@ki.se (I.S.); jan.zedenius@ki.se (J.Z.); 4Department of Molecular Medicine and Surgery, Karolinska Institutet, 17176 Stockholm, Sweden

**Keywords:** *TERT*, promoter mutation, thyroid cancer, FT-UMP, prognosis, clinical screening

## Abstract

Mutations of the *Telomerase reverse transcriptase* (*TERT*) gene promoter are recurrently found in follicular thyroid carcinoma (FTC) and follicular tumors of uncertain malignant potential (FT-UMP), but nearly never in follicular thyroid adenoma (FTA). We, therefore, believe these mutations could signify malignant potential. At our department, postoperative *TERT* promoter mutational testing of FT-UMPs was implemented in 2014, with a positive mutation screening leading to vigilant follow-up and sometimes adjuvant treatment. To date, we screened 51 FT-UMPs and compared outcomes to 40 minimally invasive FTCs (miFTCs) with known *TERT* genotypes. Eight FT-UMPs (16%) displayed *TERT* promoter mutations, of which four cases underwent a completion lobectomy at the discretion of the patient, and a single patient also opted in for radioiodine (RAI) treatment. Three mutation-positive patients developed distant metastases, registered in one patient receiving a completion lobectomy and in two patients with no additional treatment. Three out of four patients who received additional surgery, including the RAI-treated patient, are still without metastatic disease. We conclude that FT-UMPs with *TERT* promoter mutations harbor malignant potential and exhibit at least similar recurrence rates to *TERT*-promoter-mutated miFTCs. Mutational screening should constitute a cornerstone analysis in the histopathological work-up of FT-UMPs.

## 1. Introduction

Follicular thyroid tumors constitute the most commonly found thyroid neoplasia in clinical practice. The preoperative diagnosis is based on fine needle aspiration biopsy indicating the occurrence of a follicular tumor, but the final diagnosis is unfortunately not yet possible to obtain without histopathological examinations and, thus, patients are initially recommended a diagnostic lobectomy. The diagnosis of follicular thyroid carcinoma (FTC) is based on the identification of demonstrable malignant behavior, namely, capsular and/or vascular invasion, and lesions with an unequivocal absence of these criteria are diagnosed as follicular thyroid adenoma (FTA) [1]. If the tumor exhibits an ambiguous relation to either the surrounding capsule and/or adjacent blood vessels in which clear-cut invasion cannot be ruled in with absolute certainty, the lesion is considered a follicular tumor of unknown malignant potential (FT-UMP) according to the 2017 World Health Organization (WHO) criteria [1]. These criteria are prone to subjectivity and display interobserver variability, which adds to the overall uncertainty of these lesions [2]. Given these parameters, FT-UMPs demand careful histological examination of the capsule, and large parts of the gross tumor material should be submitted for histological examination, although the prognostic value of this maneuver was questioned as of late [3]. According to Swedish national guidelines, if no demonstrable invasive properties can be ascertained, FTA and FT-UMP patients are discharged with no additional follow-up, as the recurrence rates are very low. Even so, subsets of patients with FT-UMPs do recur with metastatic FTC, indicating that a fraction of FT-UMPs harbor malignant potential not yet demonstrable through conventional histological examination [1,4]. 

Numerous efforts were made to elucidate the genetic alterations discriminating between FTAs and FTCs, including tumoral messenger RNA (mRNA), microRNA, and protein expression profiles, in addition to the mutational status of various gene candidates [5,6,7,8,9,10]. One particularly promising example of the latter is recurrent promoter mutations of the *Telomerase reverse transcriptase* (*TERT*) gene, encoding the catalytic subunit of telomerase. These substitution mutations (commonly referred to as C228T and C250T) are to date the most common non-coding mutational events in human cancer, and the mutations alter the affinity for various transcription factors and increase *TERT* gene output [11,12,13]. The bulk of malignant tumors are dependent of telomerase activity to sustain proliferation without being forced into senescence due to successive telomere shortening and subsequent chromosomal impairment. Therefore, *TERT* promoter mutations confer the tumor with a selective advantage, and are recurrently observed in human cancers, often in cases with dismal outcomes [14]. *TERT* promoter mutations are observed in subsets of well-differentiated thyroid carcinomas, with advanced tumor stage and overall poor prognosis, and are regularly found in the majority of highly aggressive thyroid carcinomas (poorly differentiated thyroid carcinoma; PDTC, and anaplastic thyroid carcinoma, ATC) [15,16,17,18,19,20,21,22,23]. Considering the above, *TERT* promoter mutational screening of a follicular cell-derived thyroid carcinoma specimen is a highly specific method for the proper identification of cases at risk of future recurrences, albeit displaying a low sensitivity in well-differentiated lesions.

For follicular thyroid tumors in particular, screening for *TERT* promoter mutations was shown to be of clinical use as a diagnostic discriminator [4,19]. Indeed, retrospective mutational screening of various thyroid lesions that later recurred with distant metastases pinpointed the *TERT* promoter mutation C228T in all cases, demonstrating the rule-in properties of this marker in the clinical context [4]. At our department, *TERT* promoter mutational screening for FT-UMPs was introduced in 2014 as a standardized part of the pathology work-up in the diagnostic process, and the occurrence of a mutational event was noted in the patient charts, with each case discussed at a weekly, multi-disciplinary tumor board conference. Patients with mutation-positive tumors were given the choice of additional treatment, and the decision was made between the treating physician and the patients. We here present the clinical consequences of this screening methodology, as well as overall outcomes of this unique patient cohort.

## 2. Results

### 2.1. Clinical Characteristics and TERT Promoter Mutational Status of the Study Cohort

Summarized clinical information of the patients diagnosed with FT-UMP and comparison to the minimally invasive FTC (miFTC) cohort is presented in Table 1. The FT-UMP patients displayed a mean age of 52 years (range 15–83 years) at diagnosis and 76% were female. The mean follow-up time was 26 months (range 1–77 months). In total, eight of 51 cases showed a *TERT* promoter mutation (16%). Of the eight cases with a mutation, seven harbored the C228T and one the C250T mutation. Table 2 summarizes the findings for these eight patients. Two *TERT* promoter wild-type FT-UMPs were lost to follow-up.

### 2.2. Individual Treatments for the FT-UMPs

In total, 46 cases were diagnosed through lobectomy, while five patients underwent total thyroidectomy. The indications for an up-front total thyroidectomy in cases with Bethesda III/IV lesions included concurrent Graves’ disease (*n* = 2), bilateral multinodular goiter (*n* = 1), preoperatively verified papillary thyroid carcinoma (PTC) on the contralateral side (*n* = 1), and a follicular tumor with an augmented Ki-67 proliferation index at 10% preoperatively (*n* = 1). The treatment groups for the FT-UMPs are outlined in Figure 1. At the discretion of the patients, four opted in for additional treatment; three received a completion lobectomy but subsequently opted out for adjuvant radioiodine (RAI) ablation therapy, whereas one patient received both a completion lobectomy and RAI ablation (1.1 GBq). Four patients did not receive adjuvant therapy following diagnostic lobectomy. Among the eight patients with a mutation, three cases (38%) progressed to metastatic FTC (bone metastases *n* = 2, and lung metastasis *n* = 1), whereas none of the 41 non-mutated cases with available follow-up showed signs of relapse. Of the three patients with progression, two had no adjuvant therapy and one had a completion lobectomy but no subsequent RAI ablation. Among the five mutation-positive patients without progression, the sole patient receiving both additional surgery and RAI ablation was without recurrence after 15 months follow-up based on normal thyroglobulin levels. Of the two patients who underwent a completion lobectomy without subsequent RAI ablation, one patient was followed up with normal thyroglobulin levels, and one patient had a negative iodine scintigraphy performed. Of the two patients that opted out from additional therapy, one had a negative neck sonography and chest X-ray and the other patient is now planned for completion surgery and RAI ablation therapy.

### 2.3. Clinical Outcome in *TERT*-Promoter-Mutated FT-UMP

A Kaplan-Meier curve was used to plot the disease-free survival in *TERT*-promoter-mutated FT-UMPs and *TERT* promoter wild-type FT-UMPs. Despite the relatively short follow-up, the *TERT* promoter wild-type FT-UMPs showed prolonged disease-free survival compared to the *TERT*-promoter-mutated FT-UMPs (*p* = 0.016; Figure 2A). To compare outcomes with established malignant follicular thyroid tumors, we hypothesized that miFTCs should constitute the single most relevant tumor type for comparisons with FT-UMPs, as the former tumor exhibits low recurrence rates and demonstrates limited capsular invasion. We, therefore, included data from an miFTC cohort previously characterized for the presence of *TERT* promoter mutations, and found no difference in disease-free survival between *TERT*-promoter-mutated FT-UMPs and *TERT*-promoter-mutated miFTCs (*p* = 0.2484; Figure 2B) [19]. Furthermore, the disease-free survival was longer in *TERT* promoter wild-type miFTCs compared to *TERT*-promoter-mutated FT-UMPs (*p* < 0.0001; Figure 2B). There were too few events to perform a multivariate regression analysis. Collectively, these data suggest that *TERT*-promoter-mutated FT-UMPs should be considered malignant tumors with a higher recurrence rate than miFTCs without *TERT* promoter mutations, but similar to that of *TERT*-promoter-mutated miFTCs. 

### 2.4. Correlation to Clinicopathological Parameters in *TERT*-Promoter-Mutated FT-UMPs

When assessing differences between *TERT*-promoter-mutated and wild-type FT-UMPs, a statistically significant association was seen between the presence of a *TERT* promoter mutation and the occurrence of distant metastasis (*p* = 0.003; Table 3), higher age at surgery (both for continuous and ≥55 years, *p* = 0.016 and *p* = 0.019 respectively; Table 3), larger tumor size (*p* = 0.045, Table 3), and equivocal vascular invasion (*p* = 0.042; Table 3). There were too few events to reliably estimate regression coefficients using a multivariable logistic regression model.

## 3. Discussion

In this study, we followed FT-UMP patients as part of a clinical *TERT* promoter mutational screening program. As the denomination proposes, the malignant potential of these tumors is uncertain so far [1,4,24,25,26]. In this study, we showed that a subset of these tumors does recur with distant metastasis, and mutations in the *TERT* promoter constitute a reliable predictive marker for these events.

The advent of molecular testing partly revolutionized the possibilities for diagnostic and prognostic improvements of thyroid tumors in the clinical setting. Indeed, comprehensive multi-gene panels are rapidly gaining ground as an auxiliary methodology for pre-operative assessments of thyroid nodules [27]. As thyroid tumors sometimes pose a diagnostic challenge for the cytologist and pathologist alike, it seems inevitable that molecular testing will constitute a necessary cornerstone of the clinical work-up. One such molecular test with the potential to aid in the diagnostic process of follicular thyroid tumors is *TERT* promoter mutational screening. As a single primer pair covers the mutational hotspot region, the procedure is cheap, reliable, and fast—a stout alternative to more elaborate and time-consuming pan-gene classifiers. Moreover, several studies now highlight the rule-in ability of this molecular aberrancy in thyroid carcinoma, as the occurrence of such a mutational event strongly suggests a dismal clinical course [4,15,18,19,20,22,25]. For FT-UMPs, however, the literature regarding *TERT* promoter mutational screening outcomes is scarce (Appendix A) [4,25,28,29]. To counter this, we, therefore, employed *TERT* promoter mutational screening of FT-UMPs in a clinical setting to possibly allow for a more stringent identification of tumors with malignant properties not yet visualized by conventional histological examination. Using this maneuver, we identified eight cases with *TERT* promoter mutations out of the 51 screened tumors (16%), and three out of these eight patients since recurred with distant metastases, despite the follow-up time being short and variable. These data collectively suggest that *TERT* promoter mutational screening is a highly specific method to detect tumors with true malignant potential within the group of FT-UMPs.

The inevitable question whether these eight FT-UMPs with *TERT* promoter mutations in fact were misclassified FTCs subjected to poor tumor sampling prompted us to re-investigate these specimens from a histological perspective. No areas of clear-cut capsular or vascular invasion were noted in any sample, and the number of tissue blocks submitted for each case was judged to be adequate given the size of the primary tumor (data not shown). As full histological examination of the entire capsule with exhaustive level sectioning cannot be deemed suitable for clinical work-up in high-volume tertiary centers, our tumors are, therefore, thought to represent the current clinical situation well. Indeed, selective sampling was recommended as sufficient to detect clinically relevant malignant properties of follicular thyroid tumors [3].

The presence of a *TERT* promoter mutation was significantly associated with patient age and tumor size in our FT-UMP cohort, suggesting that this genetic event is most often reserved for large tumors resected in patients ≥55 years of age. Multivariate testing to identify independent variables was not performed due to the limited sample size, the latter being a limitation of the study design. Surprisingly, there was no association between *TERT* promoter mutations and the Ki-67 proliferation index, suggesting that low-proliferative FT-UMPs may harbor mutations as well. Moreover, equivocal vascular invasion showed a significant association with *TERT* promoter mutations, suggesting that this histological feature could pinpoint the risk of a mutation-positive FT-UMP to a better extent than when only judging tumor cell in relation to the capsule. 

When comparing mutated FT-UMPs to a previously published miFTC cohort, we could not observe any difference in outcome between *TERT*-promoter-mutated FT-UMPs and miFTCs, but a significant difference between mutated tumors (FT-UMPs/miFTCs) and wild-type miFTCs was noted (Figure 2B). Collectively, these data suggest that FT-UMPs with *TERT* promoter mutations display at least similar outcomes to their malignant counterpart, and worse than *TERT* promoter wild-type miFTCs. Therefore, *TERT*-promoter-mutated FT-UMPs should be considered an entity with malignant potential and, if significantly reproduced, in independent material, a change in nomenclature could be suggested from FT-UMPs to FT-MPs, i.e., “follicular tumors with malignant potential”. This would allow a more stringent stratification of these lesions, as they cannot be termed miFTCs given the current, gold-standard inclusion criteria of demonstrable invasive behavior [1].

The varying post-operative work-up of FT-UMPs with *TERT* promoter mutation in this cohort reflects the fact that, until now, there was no strong evidence of how to choose optimal treatment and surveillance, and the patients were as a consequence offered somewhat inconsistent treatment and follow-up options, which was not least influenced by the preference of the individual patient. Following the observations of some patients developing distant metastases, we currently offer treatment equivalent to that of malignant tumors for *TERT*-promoter-mutated FT-UMPs at our institution.

We conclude that postoperative *TERT* promoter mutational screening in clinical routine allowed us to pinpoint FT-UMP cases of which a significant subset will subsequently recur with distant metastases. Therefore, this methodology should be considered for all cases of FT-UMPs, and the presence of a mutation should justify a nomenclature change and possibly also the addition of adjuvant treatment, as the prognoses of *TERT*-promoter-mutated FT-UMPs and miFTCs are on par.

## 4. Materials and Methods 

### 4.1. Tumor Cohorts

A total of 51 FT-UMPs were included, all part of the clinical *TERT* promoter mutational screening, which started in 2014. In addition, 40 cases of minimally invasive FTCs (miFTCs) previously characterized for *TERT* promoter mutations were used as a comparison cohort when assessing recurrence rates [19]. The 51 FT-UMP cases enlisted for prospective mutational screening were collected between 2014 and early 2019. At our department, FT-UMPs make up less than 5% of all follicular thyroid tumors diagnosed yearly. All cases included in this study were diagnosed by three endocrine pathologists (Martin Hysek, Anders Höög and Carl Christofer Juhlin) and all cases were successfully interrogated for the C228T and C250T *TERT* promoter mutations using genomic DNA extracted from formalin-fixated paraffin-embedded (FFPE) tumor material. In total, 46 FT-UMPs were diagnosed as part of the primary work-up at Karolinska, and an additional five cases were included as second-opinion consultations from various departments outside of Karolinska University Hospital. Four cases showed a simultaneously occurring papillary thyroid carcinoma (PTC). Clinical data for all patients were collected from the patients’ medical records. The clinical variables included age at surgery, gender, Ki-67 proliferation index, tumor size, type of surgery, and, in certain cases, radioiodine treatment and dose, as well as the eventual time to disease recurrence and disease-specific deaths. For subsets of the cases, clinical data were previously published [4,19]. Ethical approval (EPN 2015 959-31) was obtained from the Swedish Ethical Review Authority and patient consent was granted.

### 4.2. Histopathological Criteria 

The diagnostic criteria for FT-UMP followed the recommendations briefly presented by the 2004 World Health Organization (WHO) classification of endocrine tumors which were later finalized as a defined diagnosis in 2017 [1,24]. Prior to the 2017 WHO classification, these tumors were entitled “atypical follicular thyroid adenomas” (AFTAs), and the nomenclature was changed to FT-UMP in 2017. As the diagnostic criteria are identical, we chose the term FT-UMP over AFTA to adhere to the most updated terminology. Three endocrine pathologists assessed all FT-UMPs in the cohort, and the inclusion criterion consisted of a circumscribed follicular-patterned tumor with absence of PTC-associated changes, exhibiting an equivocal relationship to the surrounding capsule and/or blood vessels [1]. The uncertainty regarding the capsular layer consisted of one or several areas in which tumor cells bulged into the capsule and in which the suspicion of complete trans-capsular invasion could not be rejected even after cutting of additional levels. Dubious relations to capsular blood vessels were defined as areas in which vascular invasion could neither be ruled in nor out. In many instances, these areas were additionally investigated using clinical routine cluster of differentiation 34 (CD34) immunohistochemistry, an endothelial marker. If the entire capsule was not originally submitted for histology, additional blocks with tumor tissue were obtained if the preceding slides indicated suspicion for an FT-UMP diagnosis. The tumors were additionally stained for Ki-67 using a clinically accredited protocol, and a proliferation index was obtained.

The 40 miFTCs used as a comparison cohort were all diagnosed as follicular thyroid tumors with capsular invasion in absence of vascular invasion according to the most recent WHO guidelines [1].

### 4.3. TERT Promoter Mutation Analysis

*TERT* promoter mutation analyses were performed on genomic DNA extracted from FFPE material. Prior to analyses, FFPE blocks were sectioned in serial consecutive μm sections and mounted on slides (for macro-dissection if desired). First and last slides prepared from each block were subjected to corresponding routine hematoxylin and eosin (H&E) staining. Two responsible endocrine pathologists (Anders Höög and Carl Christofer Juhlin) evaluated these slides to verify that representative tumor content was >70%. Genomic DNA was extracted automatically on the AS2000 Maxwell®16 MX3031 instrument using the Maxwell®16 FFPE Plus LEV DNA Purification Kit. Promega, Madison, WI, USA. The quantity and quality of the genomic DNA were measured using Nanodrop (Nanodrop technologies, Wilmington, DE, USA). 

Mutational testing was performed by polymerase chain reaction (PCR) using a single primer pair targeting the upstream positions −124 (C228T) and −146 (C250T) of the *TERT* promoter region, followed by bi-directional Sanger sequencing (Genetic Analyzer 3500 Applied Biosystems, Foster City, CA 94404 USA). Mutations were called using Mutation Surveyor, SoftGenetics LLC V4.0.4, and results were confirmed using the FinchTV Sequence Alignment Software for further visual inspection of chromatograms.

### 4.4. Statistical Analyses 

Mann–Whitney U test, chi-square, and Fisher’s exact test were used to compare *TERT* promoter mutations to clinical and pathology-associated variables in the FT-UMP cohort. Kaplan–Meier survival analysis was used to plot disease-free survival among FT-UMP patients with and without *TERT* promoter mutations, and to compare disease-free survival between *TERT*-promoter-mutated FT-UMPs and miFTCs with and without mutation from a previous publication [19]. Log-rank test was used to calculate significance. Disease-free survival time was defined as time from primary surgery to radiological or histopathological evidence of distant metastases or local recurrence. A *p*-value < 0.05 was considered as significant. Statistical computations were achieved using the SPSS Statistics 25 software (IBM, Armonk, North Castle, NY, USA), and graphs were compiled using GraphPad Prism 8 software (GraphPad Software Inc, San Diego, CA, USA).

## 5. Conclusions

Clinical routine *TERT* promoter mutational screening of FT-UMPs aids in detecting relapse-prone tumors, and the finding of such a mutation could motivate adjuvant treatment modalities even in the absence of histopathological evidence of malignant behavior.

## Figures and Tables

**Figure 1 cancers-11-01443-f001:**
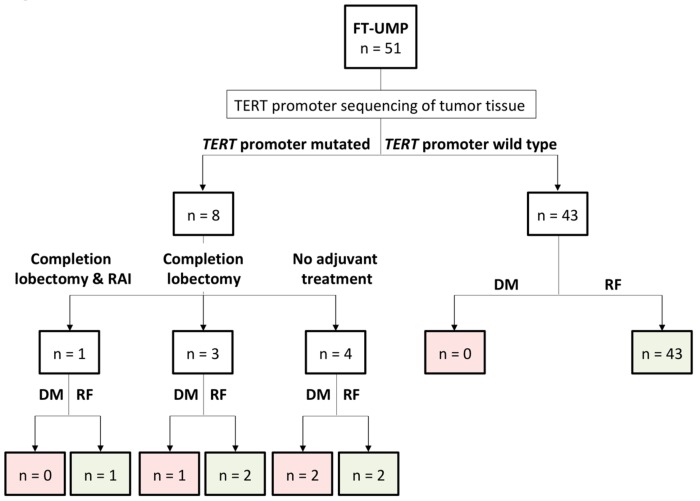
Schematic overview of the clinically recruited follicular tumor of uncertain malignant potential (FT-UMP) cohort with focus on *Telomerase reverse transcriptase* (*TERT*) promoter genotypes and patient outcome. Of the 51 clinically screened tumors, eight were found to carry a *TERT* promoter mutation, whereas the remaining 43 cases were wild type, of which two cases were lost to follow-up. While no recurrences are yet recorded among the non-mutated FT-UMP cases, three recurrences (occurrence of distant metastases) were noted among the mutated FT-UMP patients, all recorded in patients who did not receive adjuvant radioiodine therapy. DM, distant metastases; RF, recurrence-free.

**Figure 2 cancers-11-01443-f002:**
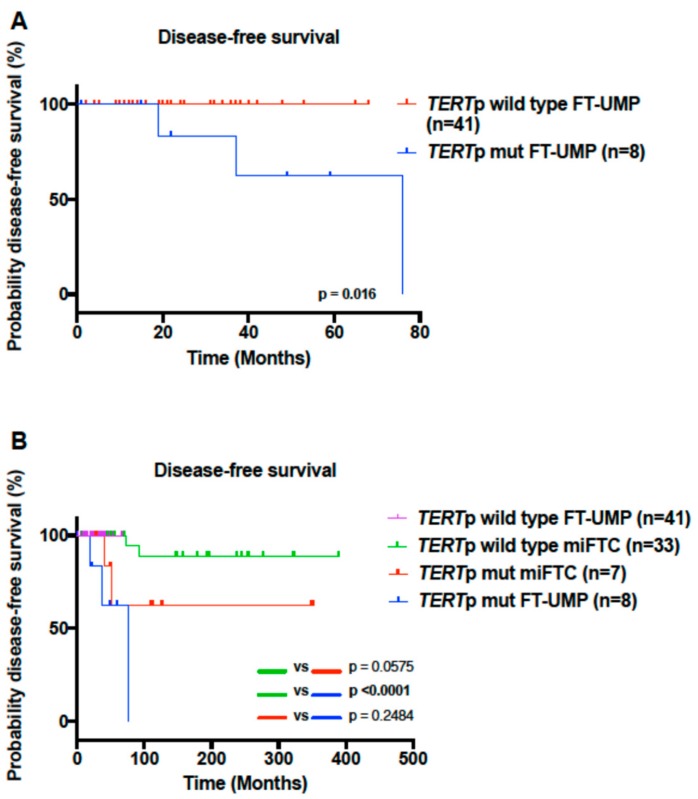
Kaplan–Meier plots illustrating disease-free survival in FT-UMPs and minimally invasive follicular thyroid carcinomas (miFTCs) stratified by *TERT* promoter (*TERT*p) genotype. (**A**) Time to detection of distant metastasis in *TERT*-promoter-mutated FT-UMP cases compared to wild-type FT-UMP cases. (**B**) Time to metastasis/recurrence in *TERT*-promoter-mutated FT-UMP cases compared to mutated and wild-type (wt) miFTC cases. The *p*-values were calculated using a log rank test. Significant *p*-values are in bold.

**Table 1 cancers-11-01443-t001:** Clinical characteristics of included tumor cases.

Parameter	FT-UMP (*n* = 51)		miFTC (*n* = 40)	
	Observation	Informative Cases	Observation	Informative Cases
Mean age at diagnosis, years	52	51	53	40
Female patients, *n*	39	51	28	40
Mean tumor diameter, mm	37	50	37	38
T category *, *n*				40
pT1	n/a		12	
pT2	n/a		15	
pT3	n/a		13	
pT4	n/a		0	
Extrathyroidal growth, *n*	n/a		1	40
Mean follow-up time, months	26	49	135	40
Outcome, *n*		49		40
Metastatic disease/recurrence	3		4	
Disease-free	46		36	
*TERT* promoter mutated, *n*	8	51	7	40

FT-UMP, follicular tumor of uncertain malignant potential; miFTC, minimally invasive follicular thyroid carcinoma; n/a, not applicable; *n*, number; *TERT*, *Telomerase reverse transcriptase*. * American Joint Commitee on Cancer (AJCC) Cancer Staging Manual 8th edition 2017.

**Table 2 cancers-11-01443-t002:** Clinical characteristics of *TERT*-promoter-mutated FT-UMP cases.

Case No.	Age at Diagnosis	Gender	Type of Surgery	RAI Treatment	Tumor Diameter (mm)	Ki-67 Index	*TERT*p Mutation	Follow-Up Time (months)	Alive	Metastatic Disease	Site of Metastasis
1	71	F	HT	No	50	7%	C250T	59	Yes	No	
2	75	F	HT + CL	No	70	6%	C228T	37	Yes	Yes	Bone
3	33	M	HT + CL	No	50	8%	C228T	49	Yes	No	
4	71	F	HT + CL	No	18	5%	C228T	22	Yes	No	
5	75	F	HT + CL	Yes	100	9%	C228T	15	Yes	No	
6	56	M	HT	No	40	5%	C228T	76	Yes	Yes	Bone
7	79	M	HT	No	No data	5%	C228T	19	Yes	Yes	Bone, lungs
8	58	F	HT	No	50	4%	C228T	1	Yes	No	

FT-UMP, follicular tumor of uncertain malignant potential; F, female; M, male; HT, hemithyroidectomy; CL, completion lobectomy; RAI, radioiodine; *TERT*p, *TERT* promoter.

**Table 3 cancers-11-01443-t003:** Association with clinicopathological parameters and *TERT* promoter mutation in FT-UMPs.

Parameter	*TERT*p-Mutated FT-UMP (*n* = 8)	*TERT*p wt FT-UMP (*n* = 43)	*p*-Value
	Observation	Observation	
Mean age at diagnosis (years)	65	50	*p* = 0.016
Age ≥55 years	7	17	*p* = 0.019
Female patients	5	34	*p* = 0.372
Mean tumor diameter (mm)	54	34	*p* = 0.045
Equivocal capsular invasion #	8	38	*p* = 0.580
Equivocal vascular invasion #	3	3	*p* = 0.042
Hypercellularity *	6	25	*p* = 0.456
Mitotic figures (≥1 per 10 HPFs)	1	6	*p* = 1.000
Degenerative changes	3	7	*p* = 0.179
Mean Ki-67 index (%)	6%	6%	*p* = 0.468
Outcome			
Metastatic disease/recurrence	3	0	*p* = 0.003

FT-UMP, follicular tumor of uncertain malignant potential; wt, wild type; *TERT*p, *TERT* promoter; HPF, high power field. # Current FT-UMP diagnostic criteria as according to the 2017 World Health Organization (WHO) guidelines. * As a qualitative and descriptive feature retrieved from the pathology report. Significant *p*-values appear in bold.

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
