# Peer review of "Clinical Routine TERT Promoter Mutational Screening of Follicular Thyroid Tumors of Uncertain Malignant Potential (FT-UMPs): A Useful Predictor of Metastatic Disease"

_cancers, 2019, doi:10.3390/cancers11101443_

Round 1

Reviewer 1 Report

 This manuscript is of potential interest in the scientific field, but requires minor revisions before its publication.

TITLE

 I suggest changing the title, in this form it loses specificity.

INTRODUCTION

Please change "encountered" with found (line 36). Please add the reference Int. J. Mol. Sci. 2018, 19, 3944; doi:10.3390/ijms19123944.

RESULTs

 Line 84: Please specify the acronym miFTC, it has never been done before. Table 1 must be redone. Use only one value per field, for ex. "Mean age at diagnosis, years" and not "Mean age at diagnosis, years (min-max)", thus it generates confusion. The same thing for "Female patients, n" and not "Female patients, n (%)". Figure 1must be improved is out of focus. Please change "wildtype" with "wild type" throughout the text. Figure 2 as the Figure 1.

DISCUSSION

The first part of the discussion should be improved by adding bibliography. Line 157: Please change"for improved diagnostics and prognostication" with "for diagnostic and prognostic improvement".

Author Response

Reviewer 1:

“This manuscript is of potential interest in the scientific field, but requires minor revisions before its publication.”

 TITLE

“ I suggest changing the title, in this form it loses specificity.”

Reply: We agree with the referee that the title could be less generic and more result-oriented. As of this, we have changed the title from “Clinical routine TERT promoter mutational screening of follicular thyroid tumors of uncertain malignant potential (FT-UMPs)” to “Clinical routine TERT promoter mutational screening of follicular thyroid tumors of uncertain malignant potential (FT-UMPs): a useful predictor of metastatic disease”. We hope this change is in line with the intentions of the referee.

INTRODUCTION

“Please change "encountered" with found (line 36).

Reply: We have changed the word “encountered” to “found” in the first sentence of the Introduction section.

Please add the reference Int. J. Mol. Sci. 2018, 19, 3944; doi:10.3390/ijms19123944.”

Reply: We thank the referee for suggesting additional literature to be cited in the context of various markers for the proper detection of FTC, and added the suggested reference (as citation #10) regarding miR9a to the section “Numerous efforts have been made to elucidate the genetic alterations discriminating between FTAs and FTCs, including tumoral mRNA, micro-RNA and protein expression profiles in addition to mutational status of various gene candidates”.

RESULTS

“Line 84: Please specify the acronym miFTC, it has never been done before”.

Reply: We concur. The abbreviation was spelled out in the Abstract, but not in the main text. We have now specified this acronym in the Results section when it first appears.

“Table 1 must be redone. Use only one value per field, for ex. "Mean age at diagnosis, years" and not "Mean age at diagnosis, years (min-max)", thus it generates confusion. The same thing for "Female patients, n" and not "Female patients, n (%)”.

Reply: Table 1 has been revised according to the suggestions made. The same was done for Table 3. For clarity, ranges between lowest and highest observation values as well as proportions (in %) were removed from the tables.

“Figure 1 must be improved is out of focus”.

Reply: Figure 1 is rendered with the resolution set to 300 dpi as outlined by the journal’s instructions. To improve the readability of the figure, we increased the font size to 16 throughout and removed the gray-colored background in some of the boxes. The raw figure is also submitted separately as a zip file, in case the rendering is compromised when it is incorporated into the Word document.

“Please change "wildtype" with "wild type" throughout the text. Figure 2 as the Figure 1”.

Reply: The suggested change has been performed throughout the manuscript as well as in Figures 1-2.

 DISCUSSION

“The first part of the discussion should be improved by adding bibliography”.

Reply: We agree with the referee, and have added existing and novel citations to the first paragraph of the Discussion section to support our claims that FT-UMPs carry an uncertain malignant potential.

“Line 157: Please change "for improved diagnostics and prognostication" with "for diagnostic and prognostic improvement”.”

Reply: The suggested change has been carried out.

Reviewer 2 Report

In this study the authors analyze the mutational status of the Telomerase Reverse Transcriptase (TERT) promoter in 51 follicular thyroid carcunomas since TERT mutations have been previously associated to a higher malignant behavior of thyroid neoplasias.

They find TERT mutations in 8 thyroid carcinomas of uncertaun malignant potential suggesting the search of this mutation as a helpful tool in the diagnosis of thyroid neoplasias.

The study is well performed, however the findings are not novel.

Criticism

In the abstract the authors use the abbreviation FTC for thyroid carcinoma. Generally this abbreviation is used for thyroid carcinoma of the folicular histotype as the authors correctly do at the beginning of the Introduction paragraph.

It is better to modify this point in the abstract.

Author Response

Reviewer 2:

”In this study the authors analyze the mutational status of the Telomerase Reverse Transcriptase (TERT) promoter in 51 follicular thyroid carcinomas since TERT mutations have been previously associated to a higher malignant behavior of thyroid neoplasias. They find TERT mutations in 8 thyroid carcinomas of uncertain malignant potential suggesting the search of this mutation as a helpful tool in the diagnosis of thyroid neoplasias. The study is well performed, however the findings are not novel.”

Reply: We thank the referee for this comment. We wish to stress that we sequenced 51 FT-UMPs rather than carcinomas, but we also included previous results from 40 miFTCs as a control group. We concur that findings of TERT promoter mutations have been reported for FTC previously, but not for FT-UMPs except for single cases with subsequent metastatic disease has been reported previously, by others and by our own group. Moreover, clinical routine mutational testing of FT-UMPs accompanied by adjustment to the treatment protocols in case of a positive finding has never has been described previously. We therefore believe our results are novel and of great clinical interest. We have added a supplementary table to the revised version, in which other studies regarding TERT promoter mutational testing of FT-UMPs (previously entitled “AFTA”) have been reported, to clarify the rarity of these reports.

Criticism:

“In the abstract the authors use the abbreviation FTC for thyroid carcinoma. Generally this abbreviation is used for thyroid carcinoma of the follicular histotype as the authors correctly do at the beginning of the Introduction paragraph. It is better to modify this point in the abstract.”

Reply: As recommended, this sentence has now been modified to “Mutations of the Telomerase reverse transcriptase (TERT) gene promoter are recurrently found in follicular thyroid carcinoma (FTC) and follicular tumors of uncertain malignant potential (FT-UMP), but nearly never in follicular thyroid adenoma (FTA)”.

Reviewer 3 Report

The information of the abstract should be more simplified and structured to give straight forward information. According to Swedish national guideline; IF no invasive properties can be ascertained, FTAs and FT-UMP patients are discharged without additional follow-up. Why do some FT-UMP patients receive surgery or even RAI? Please explain the condition. The follow-up period of FT-UMP and miFTC is not equal(26 months vs 135 months), which conclude” FT-UMPs with TERT promoter mutations harbor malignant potential and exhibit similar recurrence rates as TERT promoter mutated miFTCs” less persuasive.    In table 2 (Clinical characteristics of TERT promoter mutated FT-UMP cases); dose all these 8 patients receive RAI or WBS(whole-body scan)? If not, the exact number of metastasis may be underestimated.    In figure2. What is the differences in the green and red line(TERT wild type and mutant miFTC); If the difference is not significant; doesn’t TERT mutation increase invasion potential in miFTC. The author should add the fourth groupTERT wild FT-UMP to elucidate disease-free survival in FT-UMPs and miFTCs stratified by TERT promoter (TERTp) genotype.  Table3 should label the case number of both groups; since TERTp mutation group is older in age and larger tumor size; the conclusion of TERTp influence in metastatic potential may be biased. To make a conclusion, these variables should be adjusted.  Since these FFPE were examed by endo pathologist; more pathologic details should be listed as table 3.

Author Response

Reviewer 3:

“The information of the abstract should be more simplified and structured to give straight forward information.”

Reply: We thank the referee for this comment. We have revised the abstract to provide it with structure while still keeping the word limit to 200, and hope the changes made are in lines with your intentions.

“According to Swedish national guideline; IF no invasive properties can be ascertained, FTAs and FT-UMP patients are discharged without additional follow-up. Why do some FT-UMP patients receive surgery or even RAI? Please explain the condition.”

Reply: The purpose of this study was to evaluate the prognostic value of TERT promoter mutational screening in clinical routine, which to our knowledge never has been performed elsewhere for FT-UMPs specifically. All patients with FT-UMPs were discussed at weekly tumor board meetings, and patients with mutation-positive tumors were given the choice of additional treatment even though the current national guidelines dictate that patients with FT-UMPs should be discharged without additional treatment. The decision was made between the treating physician and the patients, and the patients were informed that this was a deliberate deviation from the current guidelines. As of such, half of the patients with mutation-positive tumors opted in for some sort of adjuvant treatment. We have now clarified this fact in the Introduction and Discussion sections.

“The follow-up period of FT-UMP and miFTC is not equal (26 months vs 135 months), which conclude” FT-UMPs with TERT promoter mutations harbor malignant potential and exhibit similar recurrence rates as TERT promoter mutated miFTCs” less persuasive.”

Reply: We agree that the follow-up time differ, as the FT-UMPs were screened quite recently, compared to our research cohort of miFTCs screened much earlier. Even so, the number of recurrent events in the FT-UMP group cannot get fewer with time, and therefore we modified the sentence in the Abstract to: “We conclude that FT-UMPs with TERT promoter mutations harbor malignant potential and exhibit at least similar recurrence rates as TERT promoter mutated miFTCs”. We also made slight modifications to the Discussion section.

“In table 2 (Clinical characteristics of TERT promoter mutated FT-UMP cases); dose all these 8 patients receive RAI or WBS(whole-body scan)? If not, the exact number of metastasis may be underestimated.”

Reply: We thank the referee for this excellent question. We have characterized five patients with mutation-positive tumors negative for metastases. The sole patient receiving both additional surgery and RAI ablation was without recurrence after 15 months follow-up based on normal thyroglobulin levels. Of the two patients who underwent a completion lobectomy without subsequent RAI ablation, one patient was followed-up with normal thyroglobulin levels, and one patient had a negative iodine-scintigraphy performed. Of the two patients that opted out from additional therapy, one had a negative neck sonography and chest X-ray and the other patient is now planned for completion surgery and RAI ablation therapy. The risk of an underestimation of metastases is therefore low. This information has now been added to the Results section.

“In figure2. What is the differences in the green and red line (TERT wild type and mutant miFTC); If the difference is not significant; doesn’t TERT mutation increase invasion potential in miFTC. The author should add the fourth group TERT wild FT-UMP to elucidate disease-free survival in FT-UMPs and miFTCs stratified by TERT promoter (TERTp) genotype.”

Reply: There is a statistically significant difference between the green and red lines (TERT wild type and mutated miFTC respectively) in Figure 2, however we did not include it in the figure as the difference between the blue (TERT promoter mutated FT-UMPs) and green lines (TERT promoter wild type miFTCs) was significant, but the difference between the blue and red lines were not. To clarify this, we have added the p value for the green and red line comparison to the figure. So yes, the TERT promoter mutations seem to confer a decrease in disease-specific survival for miFTCs, which is in accordance with previous literature.

In addition, we added the fourth group (TERT promoter wild type FT-UMPs) to the figure as suggested. However, since the group consists of patients with short follow-up time compared to the miFTC cohort, in addition to no recurrent events, it is denoted as a short and straight line that might be somewhat hard to detect in the figure.

“Table3 should label the case number of both groups; since TERTp mutation group is older in age and larger tumor size; the conclusion of TERTp influence in metastatic potential may be biased. To make a conclusion, these variables should be adjusted.”

Reply: Case numbers for both groups were added to Table 3. We also agree that it would be optimal to adjust the outcome for older age and larger tumor size by multivariate analysis, unfortunately, as stated in the previous version of our manuscript, there were too few events to perform a reliable multivariate regression analysis. In general, 10-15 cases per arm are required to reliably estimate regression coefficients in multivariable logistic regression models (PMID: 27357163). This information has now been added to the Results section. In addition, we added a paragraph of text to the Discussion section highlighting this fact, as a limitation.

“Since these FFPE were examed by endo pathologist; more pathologic details should be listed as table 3.”

Reply: We have scrutinized the pathology reports and added the information regarding some additional histopathological parameters (equivocal capsular invasion, equivocal vascular invasion, hypercellularity, mitotic figures, degenerative changes) to Table 3, and also performed additional statistical calculations. The Results section has been updated accordingly, and as a novel significant association was discovered, we have added a paragraph of text to the Discussion section as well.

Round 2

Reviewer 2 Report

The manuscript has been improved

Reviewer 3 Report

The questions were properly addressed and answered by authors.The current version of manuscript is qualified for publication.